

# Reconstruction of super-resolution from high-resolution remote sensing images based on convolutional neural networks

Yang Liu[1], Hu Xu[2] and Xiaodong Shi[3]

[1] Cloud Computing and Big Data Institute, Henan University of Economics and Law, Zhengzhou, Henan, China
[2] Henan Key Laboratory of Ecological Environment Protection And Restoration of the Yellow River Basin, Yellow River Institute Of Hydraulic Research, Zhengzhou, Henan, China
[3] School of E-commerce and Logistics Management, Henan University of Economics and Law, Zhengzhou, China

## ABSTRACT

In this study, a novel algorithm named the Edge-enhanced Generative Adversarial Network (EGAN) is proposed to address the issues of noise corruption and edge fuzziness in the super-resolution of remote sensing images. To build upon the baseline model called Deep Blind Super-Resolution GAN (DBSR-GAN), an edge enhancement module is introduced to enhance the edge information of the images. To enlarge the receptive field of the algorithm, the Mask branch within the edge enhancement structure is further optimized. Moreover, the loss of image consistency is introduced to guide edge reconstruction, and subpixel convolution is employed for upsampling, thus resulting in sharper edge contours and more consistent stylized results. To tackle the low utilization of global information and the reconstruction of super-resolution artifacts in remote sensing images, an alternative algorithm named Nonlocal Module and Artifact Discrimination EGAN (END-GAN) is proposed. The END-GAN introduces a nonlocal module based on the EGAN in the feature extraction stage of the algorithm, enabling better utilization of the internal correlations of remote sensing images and enhancing the algorithm's capability to extract global target features. Additionally, a method discriminating artifacts is implemented to distinguish between artifacts and reals in reconstructed images. Then, the algorithm is optimized by introducing an artifact loss discrimination alongside the original loss function. Experimental comparisons on two datasets of remote sensing images, NWPUVHR-10 and UCAS-AOD, demonstrate significant improvements in the evaluation indexes when the proposed algorithm is under investigation.

Corresponding author
Hu Xu, xuhu@huel.edu.cn

## INTRODUCTION

Remote sensing involves collecting ground target data through remote sensors, enabling the acquisition of surface information on a large scale (*Di, Xinfeng & Lei, 2024*; *Fahad, 2024*). It has potential in various fields, such as geological exploration, water detection, disaster monitoring, and military applications (*Dong, Chunhui & Hanyu, 2023*; *Xinyu, Yu*

*& Lizhe, 2022*). The quality resolution of remote-sensing images is a critical characteristic directly affecting results (*Kwon et al., 2023*). However, constraints in the remote sensing and imaging systems often result in lower resolutions. Hardware-based methods can enhance resolution but are costly and face challenges in transmitting large-sized images. Software-based methods offer alternative approaches by optimizing low-quality images to transform them into reconstructed high-resolution images (*Catherine, Theresa & Andréanne, 2023*). With the advancements in artificial intelligence, super-resolution image technology has become critical in extracting more information and improving visual effects (*Jiaqi, Qi & Yueting, 2023*). It allows for obtaining remote sensing images with high-quality characteristics but at a lower cost without hardware alterations. Both hardware and software approaches also have concurrent research and application significance, but software-based methods offer much universality. The super-resolution technology of remote sensing images can also benefit from other image processing techniques (*Zhihao, Zhitong & Wei, 2022*).

The concept of super-resolution image algorithms could be traced back to 1964 when *Harris (1964)* and *Goodman (1968)* extrapolated band-limited signals by employing linear interpolation or spline functions to generate high-resolution images. However, the practical application of the approach yielded poor results, leading to a lack of attention from researchers. It was not until 1984 when *Tsai & Huang (1984)* proposed a method based on image sequences in the frequency domain that the idea of image super-resolution resurfaced. Image degradation caused by frequency aliasing during the downsampling process could be mitigated by restoring the aliased information to optimize image resolution, which inspires many researchers to begin exploring the field of reconstructing super-resolution images. Numerous algorithm-based advancements have recently been introduced in reconstructing the super-resolution of remote sensing images.

Integrating generative adversarial networks (GANs) with remote sensing images presents several challenges. Three key challenges need to be addressed in the field of super-resolution images in remote sensing (*Daning, Yu & Gang, 2021*). The first challenge is to accurately restore the rich external contours and internal textures that characterize the spatial relationships of ground objects in remote sensing images. Therefore, algorithms that can effectively preserve and restore these contours and textures are essential. The second challenge arises from the local perception characteristics of convolutional neural networks (CNNs). Available algorithms often fail to incorporate global feature information in large-scale remote scenes, leading to limited performance. As a result, there is a deficiency in utilizing global remote sensing information in some available algorithms. Third, when super-resolution images are a concern, a low-resolution (LR) input corresponds to many possible high-resolution (HR) outputs in a high-dimensional image space. The training process needs to preserve generated details while suppressing artifacts.

In the research, we address these challenges by enhancing image edge information, fully utilizing global features, and suppressing artifacts. We optimize a conventional GAN-based super-resolution algorithm commonly implemented in other domains to achieve super-resolution images in remote sensing. Thus, the EGAN is proposed as an algorithm leading to the super-resolution of remote sensing images based on the DBSR-GAN. The EGAN

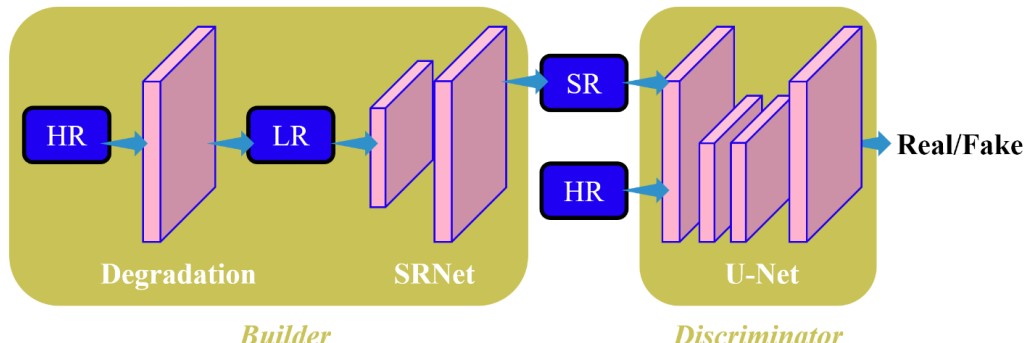

**Figure 1  Structure diagram of BSRGAN algorithm in this study.**

enhances edge details by integrating an edge enhancement network and improving the Mask branch for complex attention maps. Lcons are employed to optimize the reconstruction of edge information and replace nearest-neighbor upsampling with sub-pixel convolution. However, the EGAN lacks global spatial connectivity. To address this, nonlocal modules are incorporated before and after the feature extraction network. By dividing feature maps into a $2 \times 2$ grid, nonlocal operations are performed within each grid.

# SUPER-RESOLUTION OF REMOTE SENSING IMAGE WITH FUSED EDGE ENHANCEMENT

An improved version of the DBSRGAN algorithm is proposed to address the issues of unclear reconstruction of high-resolution image edge details in the available algorithm. Firstly, we introduce the principle and framework of the baseline algorithm. We integrate an edge enhancement network and enhance the Mask branch to build upon this foundation. An edge consistency loss is also introduced to optimize the reconstruction of edge information. Moreover, the algorithm's overall sampling method with subpixel convolution is replaced.

## Baseline algorithm model

The research adopts the deep-blind image super-resolution algorithm DBSR-GAN as our baseline algorithm. The DBSR-GAN is a degradation algorithm that addresses the challenges of deep-blind image super-resolution. It incorporates various degradation factors such as blurring, downsampling, and noise, thus utilizing a more complex and practical degradation network. The structure of the network is depicted in Fig. 1. The generated network follows a single-stage post-upsampling design, consisting of the degradation network for generating LR images with different degradation types and the SRNet to reconstruct SR images.

To distinguish between correctly classified and misclassified classes, the discriminator employs the spectral normalized U-Net algorithm (*Schonfeld, Schiele & Khoreva, 2020*) to calculate the difference between the SR and the HR images. It provides global (whole image) and local (pixel-per-pixel) feedback for the SR image.

The degradation of remote-sensing images primarily involves blurring, subsampling, and noise. We aim to expand the degradation space for these three key factors to enhance

the usefulness of degradation algorithms. The DBSR-GAN algorithm achieves this by incorporating a random shuffling strategy.

The generator network G in the DBSR-GAN adopts the enhanced super-resolution approach by utilizing the multilevel residual network algorithm from the adversarial network ESRGAN. All batch normalization (BN) layers are removed from the algorithm, which is done to prevent artifacts in the reconstructed image and enhance the generator's generalization capability, especially when training and test datasets exhibit significant differences. The algorithm employs the Residualin-ResidualDenseBlock (RRDB) as the basic block in the super-resolution network, effectively increasing the network capacity through dense connections.

Compared to conventional generative adversarial network-based algorithms, the DBSR-GAN exhibits a restoration for superior texture structure in remote sensing images, resulting in higher-quality reconstructed images. However, it still faces the challenge of uneven edges in reconstructed remote sensing images. Therefore, to obtain remote sensing images with clearer edges and richer details, the DBSR-GAN algorithm is utilized as the baseline algorithm. A super-resolution algorithm for remote sensing images called EGAN is proposed by integrating an edge enhancement network.

## EGAN algorithm

The blind image super-resolution algorithm, DBSR-GAN, is implemented as the baseline algorithm in the article. To address the challenge of super-resolution for remote sensing images, a fusion algorithm called Edge-Enhanced Generative Adversarial Network (EGAN) is proposed that incorporates an Edge Enhancement Network. The algorithm aims to achieve a four-time improvement as much as possible in reconstructing super-resolution of remote sensing images. The overall framework of the EGAN is depicted in Fig. 2. In the EGAN, the Degradation Network generates LR images, while the Super-Resolution Network (SRNet) generates an intermediate super-resolution image, denoted as $I_{Base}$. The Edge Enhancement Network (EEN) extracts edge information from $I_{Base}$ for enhancement. The enhanced edge information is then combined with the final super-resolution image, resulting in improved edge quality and reduced noise and artifacts. This final super-resolution image is denoted as $I_{HR}$. The discriminator employed in the EGAN is based on the U-Net discriminator with spectral normalization utilized in the baseline DBSR-GAN algorithm.

Some algorithms to enhance edges employ approximate filters by implementing CNNs to extract and preserve image edges (*Xu et al., 2015*; *Liu, Pan & Yang, 2016*). However, these algorithms aim to retain important structures and remove details rather than accurately estimating the edges of the reconstructed images. The integration of an EEN into the DBSR-GAN algorithm is proposed to address this issue. The network architecture consists of the Residual-in-Residual Dense Connections (RRDC) branch for edge mapping extraction and the Mask branch for adaptive learning of specific weight matrices to focus on real edge information and eliminate noise.

The SRNet generates the intermediate super-resolution image, denoted by I_Base, which is then processed by a Laplacian operator (*Kamgar & Rosenfeld, 1999*) to extract

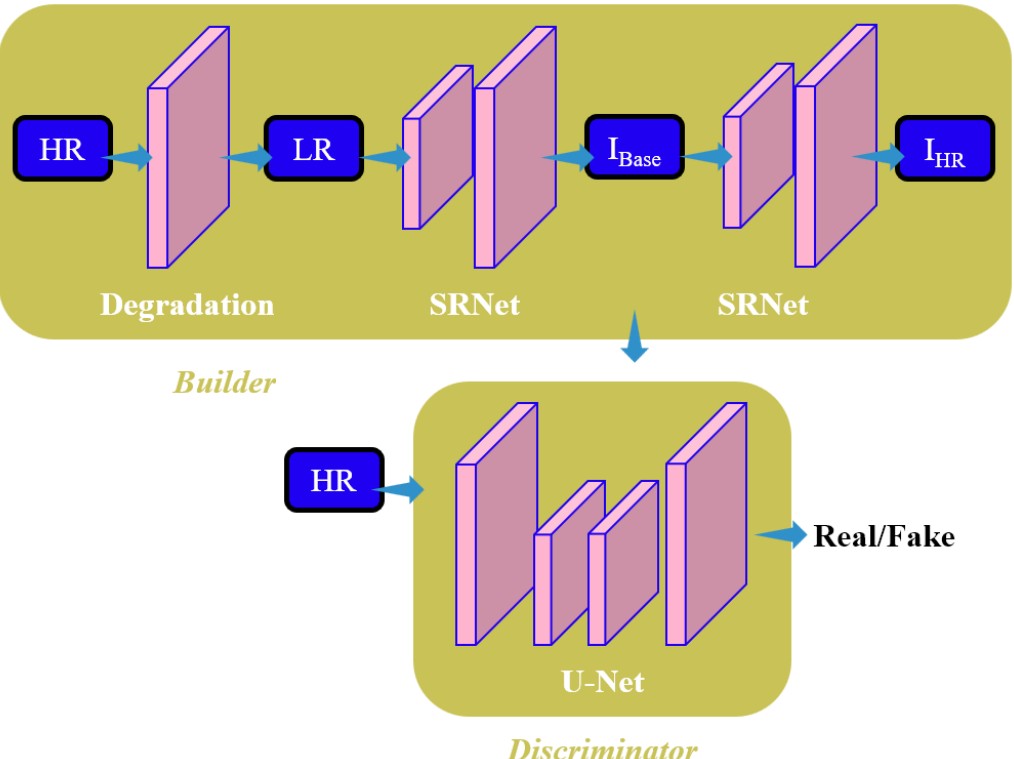

**Figure 2** Structure diagram of EGAN model in this study.

edge information. The Laplacian operator performs a second-order derivative operation on the image, producing a sharp zero-crossing point at the edges. This operation is achieved implementing a convolutional kernel of ([−1,−1,1],[−1,8,−1],[−1,−1,−1]). The edge extraction process using the Laplacian operator is represented by Eq. (1).

$$E(x,y) = L(x,y) \otimes I(x,y) \tag{1}$$

$L(x,y)$ and $E(x,y)$ represent the discrete convolution mask and the extracted edge mapping, respectively, and $\otimes$ denote the convolution operator.

The extracted edge information using the Laplacian operator is further processed with stride convolution to extract the edge mapping and convert it to the LR space. The Mask branch masters the transformed features to detect and remove false edge points generated while extracting edges. The RRDC branch consists of dense residual connection blocks (RRDB), the same as those employed in the baseline network. Upsampling operations are applied to enhance the edge mapping to the HR space. This process can be represented by Eq. (2).

$$I_{Edge}^* = S(F(D(I_{Edge})) \otimes M(D(I_{Edge}))) \tag{2}$$

where $D(\cdot)$, $F(\cdot)$, $M(\cdot)$, and $S(\cdot)$ represent the stride convolution downsampling, the operation of the dense residual connection branch, the Mask branch, and upsampling, respectively.

Finally, the enhanced edge information is substituted for the noisy edge information extracted from the intermediate super-resolution image ($I_{Base}$), resulting in the final super-resolution image ($I_{HR}$). The choice of the upsampling method in the algorithm affects the quality of reconstructed images. Considering the influence of the sub-pixel level in the image, the suggested algorithm replaces the nearest neighbor upsampling implemented in the SRNet and the EEN with sub-pixel convolution. This optimization addresses the limitations of conventional interpolation methods and yields better reconstruction results.

The original structure of the Mask branch in the edge enhancement module is relatively simple and cannot capture a larger range of information. This study further improves the Mask branch to increase the model's receptive field, and more complex attention maps are obtained. Stride convolution and sub-pixel convolution operations were employed to preserve as much detailed information as possible in the image. The stride convolution increased the receptive field size by one basic block and two additional basic blocks. The downscaled feature map was then expanded by implementing sub-pixel convolution and subsequently transformed by one basic block and a $1 \times 1$ convolutional layer to amplify the features. Finally, a sigmoid layer normalized the output scores to [0,1]. In this process, simplified residual blocks, referred to as RBBlocks, were utilized as the basic blocks.

The loss function is crucial in evaluating the disparity between the generated $I_{Base}$ or $I_{HR}$ images produced by the network model and the HR images utilized for training purposes. To achieve clearer edge contours and consistent styles, the training process of the network introduces the image consistency loss $L_{cons}$ on top of the L, pixel loss, perceptual loss Lpercep, and adversarial loss Lady. The overall loss of the model is represented by Eq. (3).

$$L_{G\_een} = \lambda_1 L_1 + \lambda_2 L_{percep} + \lambda_3 L_{adv} + \lambda_4 L_{cons} \qquad (3)$$

where $\lambda_1$, $\lambda_2$, $\lambda_3$, and $\lambda_4$ represent the weight parameters for different losses. Based on experimental results, we set $\lambda_1 = 1$, $\lambda_2 = 1$, $\lambda_3 = 0.001$, and $\lambda_4 = 5$, respectively.

The image consistency loss $L_{cons}$ is defined as the Charbonnier loss (*Lai, Huang & Ahuj, 2017*) between $I_{HR}$ and HR images. The Charbonnier loss is more stable when compared to the $L_1$ loss and can handle outliers during the training process to improve reconstruction accuracy. Additionally, to suppress potential edge distortions that may occur during the super-resolution process of remote sensing images, the research considers introducing the edge consistency loss $L_{edge\_cons}$, which calculates the Charbonnier loss between the edges $I_{edge\_HR}$ extracted from the high-resolution image and the enhanced edges $I_{edge}^{*}$. However, through experiments, it was found that $L_{edge\_cons}$ introduced varying degrees of noise. Therefore, only the image consistency loss Lcons was introduced during the training process. The above process can be represented by Eqs. (4) and (5), respectively.

$$L_{cons}(\theta_G) = \rho(HR - I_{HR}) \qquad (4)$$

$$L_{edge\_cons} = \rho(I_{edge\_HR} - I_{edge}^{*}) \qquad (5)$$

where $\rho(\cdot)$ represents the Charbonnier penalty function.

## Experiment and the results of the analysis

In the research, we conducted training and evaluation by utilizing two remote-sensing image datasets, namely, NWPUVHR-10 and UCAS-AOD. The NWPUVHR-10 dataset consists of 800 high-resolution satellite images, including 650 images with ground objects and 150 background images without objects. The UCAS-AOD dataset contains 2,420 high-resolution satellite images, including 1,510 images with objects and 910 background images. The randomly selected 160 images from each dataset set were used as the test set, and the rest were utilized for training.

Most available works for image super-resolution use bicubic downsampling to obtain LR images for training and testing. However, real-world images often suffer from complex, blurred and noise degradation, which can be represented by Eq. (6) in the degradation process.

$$I_{LR} = D(x) = (y \otimes k) \downarrow_s + n. \tag{6}$$

To comprehensively validate the superiority of the proposed algorithm in the article, we set up four different degradation types for the test sets, expressed as follows:

$$\text{Type I}: I_{LR} = D(x) = y \downarrow_s \tag{7}$$

$$\text{Type II}: I_{LR} = D(x) = (y \otimes k) \downarrow_s \tag{8}$$

$$\text{Type III}: I_{LR} = D(x) = \&[(y \otimes k) \downarrow_s + n]_{JPEG} \tag{9}$$

$$\text{Type IV}: I_{LR} = D(x) = [(y \otimes k) \downarrow_s + n]_{JPEG} \tag{10}$$

The experimental setup was conducted on a 12GB NVIDIA GeForce RTX 3080 Ti GPU, using the Pytorch framework for deep learning. Initially, a pre-trained model named Enet, consisting of a generator with $L_1$ loss and $L_{cons}$ loss, was trained. This Enet model was then used as the foundation to train the final EGAN model. The images were cropped into 320 × 320 image patches during the training process, with a magnification factor set to 4. The batch size was set to 4. The initial learning rate for the pre-training of the Enet model was 0.0001, with 100,000 iterations. The EGAN model's initial learning rate was set to 0.00005, with 100,000 iterations. The Adam optimizer was used for both models. The evaluation metrics for the effectiveness of image reconstructions included SSIM, PSNR, and Root Mean Square Error (RMSE), respectively. Higher SSIM and PSNR scores and lower RMSE scores indicate better image reconstruction quality. Additionally, subjective evaluation was performed by assessing human observers' visual perception of image quality.

To objectively analyze the performance of the proposed algorithm, four different algorithms were compared, namely, Bicubic, ESRGAN, RealSR, and DBSR-GAN, employing the UCAS-AOD and NWPUVHR-10 remote sensing image datasets. ESRGAN, RealSR, and DBSR-GAN are classical or state-of-the-art algorithms in the field of image super-resolution. The results were obtained through group testing of different algorithms, as shown in Tables 1–4.

The following conclusions can be drawn: firstly, on the Type I test set, ESR-GAN achieved the second-best PSNR result, as the training process involved bicubic degradation, but the network failed to suppress artifacts generated by adversarial training effectively. Secondly,

**Table 1  Comparison of experimental results of Type I.**

| Index | Bicubic | ESRGAN | RealSR | BSRGAN | EGAN |
|---|---|---|---|---|---|
| SSIM | 0.8 | 0.8 | 0.7 | 0.8 | 0.8 |
| PSNR | 27.5 | 26.3 | 25.6 | 25.6 | 26.8 |
| RMSE | 12.1 | 14.8 | 14.7 | 14.5 | 12.9 |

**Table 2  Comparison of experimental results of Type II.**

| Index | Bicubic | ESRGAN | RealSR | BSRGAN | EGAN |
|---|---|---|---|---|---|
| SSIM | 0.8 | 0.7 | 0.7 | 0.7 | 0.7 |
| PSNR | 25.9 | 23.9 | 24.0 | 24.1 | 25.2 |
| RMSE | 14.6 | 18.1 | 18.0 | 17.5 | 15.6 |

**Table 3  Comparison of experimental results of Type III.**

| Index | Bicubic | ESRGAN | RealSR | BSRGAN | EGAN |
|---|---|---|---|---|---|
| SSIM | 0.8 | 0.7 | 0.7 | 0.7 | 0.8 |
| PSNR | 26.4 | 24.1 | 24.3 | 25.2 | 26.4 |
| RMSE | 13.8 | 17.7 | 17.2 | 15.3 | 13.6 |

on the Types II and III test sets, the RealSR performed better than the ESR-GAN overall due to the integration of real image degradation information. Finally, on the Type IV test set, all algorithms showed significant degradation in performance. Still, the proposed EGAN algorithm, benefiting from the research's advanced methods and degradation networks, achieved the best results across all types of test sets. Furthermore, the proposed algorithm progressed in various evaluation metrics compared to the baseline DBSR-GAN algorithm on both datasets and different degradation types. For instance, in the Type IV test set, there was only a 0.066 in SSIM, a 1.372 dB in PSNRi improvements, and a 12% decrease in RMSE. These experimental results validated the effectiveness and superiority of the proposed EGAN algorithm for remote-sensing image datasets.

The Bicubic algorithm showed high objective metrics but exhibited poor subjective visual quality with blurry texture details (Fig. 3). Therefore, it was not considered as the optimal result in the comparative analysis. Figure 3 provides a visual comparison of different algorithms, indicating that the conventional Bicubic approach fails to generate additional details, the ESR-GAN tends to introduce excessive artifacts, the RealSR fails to eliminate degradation, and the DBSR-GAN does not sufficiently restore natural texture. In contrast, the proposed EGAN algorithm can effectively reconstruct the edge information of the image.

**Table 4** Comparison of experimental results of Type IV.

| Index | Bicubic | ESRGAN | RealSR | BSRGAN | EGAN |
|---|---|---|---|---|---|
| SSIM | 0.5 | 0.5 | 0.5 | 0.6 | 0.6 |
| PSNR | 23.5 | 22.2 | 22.5 | 23.1 | 24.5 |
| RMSE | 19.1 | 20.1 | 19.9 | 19.7 | 17.3 |

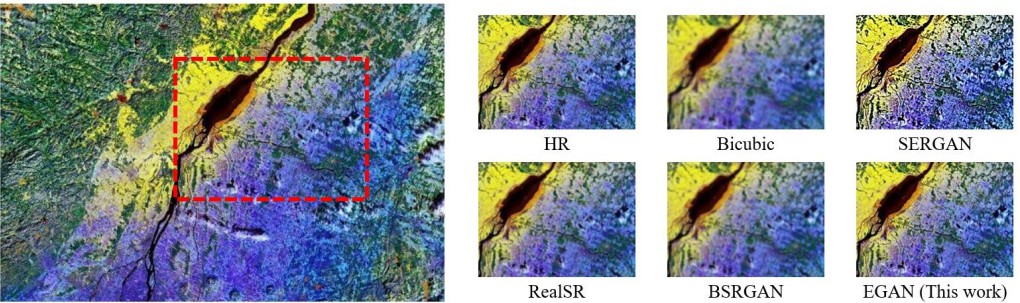

**Figure 3** Visual effect comparison for Type III.

# IDENTIFICATION OF SUPER-RESOLUTION REMOTE SENSING IMAGES BY ARTIFACTS BASED ON NONLOCAL MODULES

The available super-resolution algorithms for remote sensing images have limitations in considering global spatial connectivity and texture refinement. Additionally, GAN training is known to be unstable and can easily introduce artifacts. To address these issues, the article proposes a novel algorithm called END-GAN, which is an improved form of the EGAN algorithm to achieve super-resolution in remote sensing images. To enhance the feature extraction stage of the algorithm, two nonlocal modules are introduced that effectively combine wide-area and remote image information during the training process. By incorporating the nonlocal modules, the proposed algorithm can capture global spatial connectivity and refine texture details, improving performance in reconstructing high-resolution images. Furthermore, the proposed algorithm employs an artifact discriminant learning method, which enables the algorithm to retain true details while suppressing artifacts. Hence, this further enhances the algorithm's capability to generate high-quality outcomes with artifact-free images.

## ENDGAN algorithm
### Nonlocal module

Convolution operations typically process a local neighborhood one at a time, which can lead to limitations in capturing global dependencies. To overcome this, some algorithms have been developed based on nonlocal mean filtering (*Buades, Coll & Morel, 2005*), which searches for similar pixels globally across the entire LR image. These methods calculate weighted scores for all pixels in the image, allowing distant pixels to simultaneously contribute to a specific position's response.

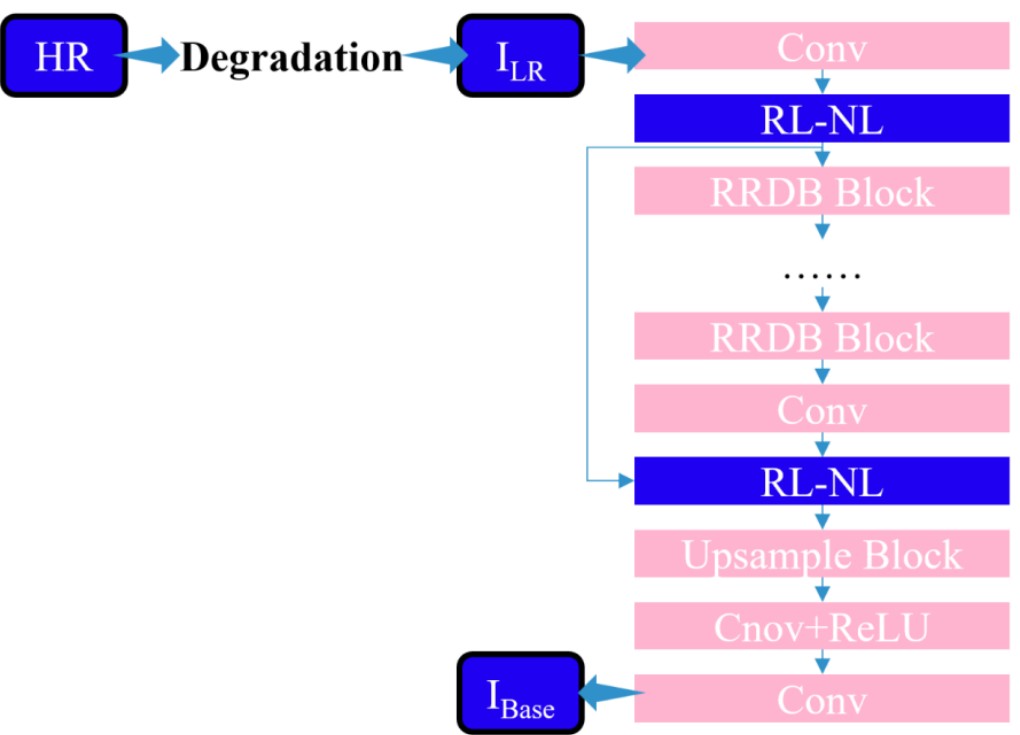

**Figure 4  Structure diagram of SR network framework of this work.**

The research proposes integrating a region-level nonlocal module (RL-NL) before and after running the SRNet, as depicted in Fig. 4. The nonlocal operations within this module can gather relevant features from the entire image. Mathematically, the nonlocal operations can be expressed by Eq. (11).

$$y_i = \frac{1}{\sum_{\forall j} f(x_i, x_j)} \sum_{\forall j} f(x_i, x_j) g(x_j) \tag{11}$$

where "i" denotes the output feature position, "j" represents all possible positions, and "x" and "y" designate inputs and outputs of the nonlocal operations, respectively. The pairwise function, $f(x_i, x_j)$, calculates the relationship between $x_i$ and $x_j$, while the function $g(x_i)$ calculates the representation at position j.

To evaluate the relationship between the two positions, an embedded Gaussian function is employed that calculates the similarity in the embedded space, as expressed in Eq. (12):

$$f(x_i, x_j) = \exp((W_\theta x_i)^T W_\phi x_j) \tag{12}$$

where $W$ denotes the weight matrix. Considering a linear embedding form, $g{:}g(x_i){=}W_g x_j$, the overall process of a nonlocal block can be represented by Eq. (13).

$$z_i = W_z y_i + x_i \tag{13}$$

where $y_i$ is given in Eq. (11), $W_z$ represents the weight matrix, and "$+x_i$" denotes the residual connection. The overall structure is illustrated in Fig. 5.

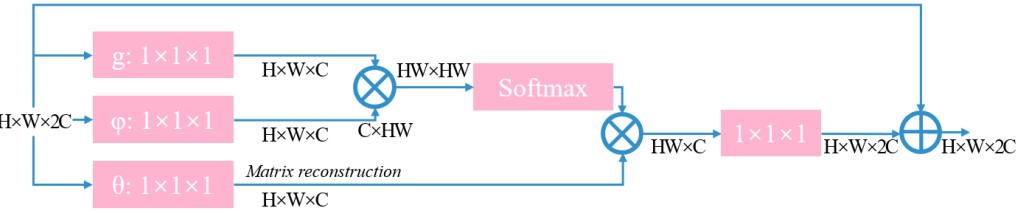

**Figure 5  Structure diagram of non-local module.**

### Artifact discriminant learning process

In image super-resolution, a LR input corresponds to many possible high-resolution (HR) outputs in a high-dimensional image space. The adversarial loss, Ladv, updates the results along multiple possible directions, leading to an unstable optimization process and resulting in artifacts and loss of details. This study introduces a method called artifact discrimination learning to preserve the true details and suppress artifacts while stabilizing the training of the generative adversarial network (GAN).

Firstly, we assume that the resolution of the input color image, $I_{SR}$, is H $\times$ W $\times$ 3, and we use $M(i,j) \in [0,1]$ to represent the probability that $I_{SR}(i,j)$ is an artifact pixel. The objective is to obtain a pixel-level mapping, $M \in R^{H \times W \times 1}$. Since both artifacts and details belong to high-frequency image components, the residual between the ground truth image, $I_{HR}$, and the reconstructed image, $I_{SR}$, is first computed to extract the high-frequency components represented by Eq. (14).

$$R = I_{HR} - I_{SR}. \tag{14}$$

As restoring fine texture information in the super-resolution process of images is more challenging, artifacts are often composed of overshoot pixel values. Therefore, the local variance of the residual map, R, is computed as the main indicator of artifact pixels. The calculation process is represented by Eq. (15).

$$M(i,j) = \mathrm{var}\left(R\left(i - \frac{n-1}{2} : i + \frac{n-1}{2}, j - \frac{n-1}{2} : j + \frac{n-1}{2}\right)\right) \tag{15}$$

where var represents the variance operator and n denotes the local window size and is set to 7.

Since the local variance is calculated with a very small receptive field, distinguishing artifacts from edges and textures can be unstable and may result in incorrect penalization of genuine details. To address this issue, a stable patch-level variance is computed based on the entire residual map, R, represented by Eq. (16). Typically, images with smoother regions have a smaller $\sigma$ value when compared to images with rich texture details. By scaling the main mapping, M, by $\sigma \cdot$ M, a more reliable artifact mapping can be obtained and denoted by

$$\sigma \& = (\mathrm{var}(R))^{\frac{1}{q}}. \tag{16}$$

Furthermore, to further stabilize the training process and refine the artifact map, a dynamic gradient descent optimization is employed for the image super-resolution model,

represented by $\Psi$. The exponential moving average (EMA) method is used to obtain a more stable model, $\Psi_{EMA}$, from $\Psi$, as shown in Eq. (17).

$$\Psi_{EMA}^{(k)} \& = \alpha \cdot \Psi_{EMA}^{(k-1)} + (1-\alpha) \cdot \Psi^{(k)} \qquad (17)$$

where $\alpha$ represents a weighting parameter set to 0.999. $(\cdot)^{1/q}$ scales the global variance, var(R), to an appropriate proportion, and in our experiments, $q = 5$ is set. The outputs of the two super-resolution models as $I_{SR1} = \Psi(I_{LR})$ and $I_{SR2} = \Psi_{EMA}(I_{LR})$ are denoted. Two residual maps, $R_1 = I_{HR} - I_{SR2}$ and $R_2 = I_{HR} - I_{DR2}$ are calculated and the artifact map, $\sigma \cdot M$, is defined by using Eq. (18).

$$M_{refine}(i,j) = \begin{cases} 0, if \left|R_1(i,j)\right| < \left|R_2(i,j)\right| \\ \sigma \cdot M(i,j), if \left|R_1(i,j)\right| \geq \left|R_2(i,j)\right| \end{cases} \qquad (18)$$

The refined artifact mapping, $M_{refine}$, will only penalize locations where $|R_1(i,j)| \geq |R_2(i,j)|$. In locations where the residual of $I_{SR1}$ is smaller than that of $I_{SR2}$, the model, $\Psi$, updates in the correct direction and is not affected by the refinement process.

After the aforementioned calculations, the refined artifact map, $M_{refine}$, is obtained and the artifact discrimination loss, $L_{artif}$, is defined by Eq. (19).

$$L_{artif} = \left\| M_{refine} \cdot (I_{HR} - I_{SR1}) \right\|_1. \qquad (19)$$

By introducing the artifact discrimination loss, $L_{artif}$, into the EGAN baseline model, the final loss function is represented by Eq. (20).

$$L_G = L_{G_{-een}} + \beta L_{artif} \qquad (20)$$

where $\beta$ represents a weighting parameter set to 5 based on experimental results. The process of artifact discrimination learning is as follows: $I_{LR}$ is fed into both the super-resolution models, $\Psi$ and $\Psi_{EMA}$, resulting in outputs $I_{SR1}$ and $I_{SR2}$, respectively. Then, the original high-resolution image, $I_{HR}$, is used along with $I_{SR1}$ and $I_{SR2}$ to construct the refined artifact map, $M_{refine}$. The artifact discrimination loss, $L_{artif}$, is computed based on $I_{HR}$, $I_{SR1}$, and $M_{refine}$. Finally, the overall model loss, $L_G$, is used to optimize the model, $\Psi$, and the parameters of $\Psi$ are aggregated over time to form $\Psi_{EMA}$. This process is repeated until the algorithm converges.

## Experiment and the analysis of results

The proposed algorithm utilizes the same architecture and training methods as the EGAN algorithm discussed in 'Super-resolution of Remote Sensing Image with Fused Edge Enhancement'. Specifically, the training and testing phases were conducted by using Nvidia GTX 3080Ti GPU and the Pytorch deep learning platform. Two remote-sensing image datasets, NWPUVHR-10 and UCAS-AOD, were implemented for training and testing. In addition to the objective and subjective evaluations employed in the previous section, the mean average precision (mAP) of object detection results is also considered, as remote sensing images are often implemented for ground target detection tasks.

To better evaluate the performance of the proposed END-GAN algorithm, six algorithms are compared, namely, Bicubic, ESRGAN, RealSR, Real-ESRGAN, DBSR-GAN, and EGAN

**Table 5 Comparison of experimental results of Type I.**

| Index | Bicubic | ESRGAN | RealSR | BSRGAN | BSRGAN | EGAN This work | ENDGAN This work |
|-------|---------|--------|--------|--------|--------|----------------|------------------|
| SSIM  | 0.8  | 0.8  | 0.8  | 0.8  | 0.8  | 0.8  | 0.8  |
| PSNR  | 27.5 | 26.3 | 25.6 | 25.8 | 25.6 | 26.8 | 27.4 |
| RMSE  | 12.1 | 14.8 | 14.7 | 14.3 | 14.5 | 129  | 12.0 |

**Table 6 Comparison of experimental results of Type II.**

| Index | Bicubic | ESRGAN | RealSR | BSRGAN | BSRGAN | EGAN This work | ENDGAN This work |
|-------|---------|--------|--------|--------|--------|----------------|------------------|
| SSIM  | 0.8  | 0.1  | 0.7  | 0.7  | 0.7  | 0.7  | 0.7  |
| PSNR  | 25.9 | 23.9 | 24.0 | 24.5 | 24.1 | 25.2 | 26.4 |
| RMSE  | 14.6 | 18.0 | 18.0 | 17.1 | 17.5 | 15.7 | 14.7 |

**Table 7 Comparison of experimental results of Type III.**

| Index | Bicubic | ESRGAN | RealSR | BSRGAN | BSRGAN | EGAN This work | ENDGAN This work |
|-------|---------|--------|--------|--------|--------|----------------|------------------|
| SSIM  | 0.8  | 0.7  | 0.7  | 0.7  | 0.7  | 0.8  | 0.8  |
| PSNR  | 26.4 | 24.1 | 24.3 | 25.4 | 25.2 | 26.4 | 26.9 |
| RMSE  | 13.8 | 17.7 | 17.2 | 15.3 | 15.3 | 13.6 | 12.7 |

(the proposed algorithm) by employing the two remote sensing image datasets. The results are shown in Tables 5–8.

The Real-ESRGAN achieved suboptimal results on most test sets due to higher-order degradation models and using Sinc filters to simulate common ringing and overshoot artifacts. On the other hand, the END-GAN algorithm, benefiting from the improvement strategies proposed in the article, achieved the best results on different test sets. Furthermore, compared to the proposed EGAN algorithm in 'Super-resolution of Remote Sensing Image with Fused Edge Enhancement', improvements in multiple evaluation metrics for different degradation types are attained. When Type I is taken as an example, the END-GAN model showed a 0.536 dB improvement in PSNR, around 0.02 improvement in SSIM, and around 7% reduction in RMSE compared to the EGAN algorithm.

## CONCLUSION

The research addresses the challenges and difficulties in the field of super-resolution by examining available image super-resolution algorithms. Solutions are provided to overcome three specific problems. Firstly, to address the issue of blurred edge details in high-resolution image reconstruction, a super-resolution reconstruction algorithm called EGAN is proposed, which integrates an edge enhancement network into the DBSR-GAN algorithm to extract and enhance the original edge information. Additionally, improvements are made to the Mask branch and the upsampling method. Experimental

**Table 8  Comparison of experimental results of Type IV.**

| Index | Bicubic | ESRGAN | RealSR | BSRGAN | BSRGAN | EGAN This work | ENDGAN This work |
|---|---|---|---|---|---|---|---|
| SSIM | 0.5 | 0.5 | 0.5 | 0.6 | 0.5 | 0.6 | 0.6 |
| PSNR | 23.5 | 22.2 | 22.5 | 23.2 | 23.1 | 24.5 | 24.9 |
| RMSE | 19.1 | 20.1 | 19.9 | 19.6 | 19.7 | 17.3 | 17.2 |

results on two remote sensing image datasets validate the feasibility and superiority of the proposed method. Secondly, the limitations of the EGAN model are analyzed. Nonlocal modules are introduced in the feature extraction stage to improve the utilization of global information and the limitations of local feature extraction. This enables effective integration of wide-area and remote image information, enhancing the algorithm's capability to extract global target features. Regional nonlocal operations are applied by dividing the feature map into $2 \times 2$ grids to reduce the computational burden. Lastly, to address the issues of unstable training of GANs and artifacts in reconstructed images, the END-GAN algorithm is proposed. It utilizes artifact discrimination to suppress artifacts while retaining real details during network training. An artifact discrimination loss is introduced to optimize the algorithm. Experimental results demonstrate improvements in various evaluation indexes, and the reconstructed images exhibit better detection results when compared to the original images.

### Funding

This work was supported by the Henan Provincial Smart Teaching Research Project of Higher Education and the Henan Provincial Science and Technology Key Project Foundation (No. 212102310085, 222102210142, 222102210252), the Key Research Project of Colleges and Universities in Henan Province of China (No. 23A520055), and the Henan Provincial Key Laboratory of Ecological Environment Protection and Restoration of the Yellow River Basin Open Research Fund (No. LYBEPR202202). The funders had no role in study design, data collection and analysis, decision to publish, or preparation  of the manuscript.

### Grant Disclosures

The following grant information was disclosed by the authors:
Henan Provincial Smart Teaching Research Project of Higher Education.
Henan Provincial Science and Technology Key Project Foundation: 212102310085, 222102210142, 222102210252.
Key Research Project of Colleges and Universities in Henan Province: 23A520055.
Henan Provincial Key Laboratory of Ecological Environment Protection and Restoration of the Yellow River Basin Open Research Fund: LYBEPR202202.

## Competing Interests

The authors declare that they have no competing interests.

## Author Contributions

- Yang Liu conceived and designed the experiments, performed the experiments, analyzed the data, performed the computation work, prepared figures and/or tables, authored or reviewed drafts of the article, and approved the final draft.
- Hu Xu conceived and designed the experiments, performed the experiments, analyzed the data, performed the computation work, prepared figures and/or tables, authored or reviewed drafts of the article, and approved the final draft.
- Xiaodong Shi conceived and designed the experiments, performed the experiments, analyzed the data, performed the computation work, prepared figures and/or tables, authored or reviewed drafts of the article, and approved the final draft.

## Data Availability

The code is available in the Supplementary File.

The NWPU VHR-10 dataset and SSDD dataset according to the standard coco dataset are available at Github: https://github.com/chaozhong2010/VHR-10_dataset_coco.

The UCAS-AOD remote sensing image data set is available at https://hyper.ai/datasets/5419.

## Supplemental Information

Supplemental information for this article can be found online at http://dx.doi.org/10.7717/peerj-cs.2218#supplemental-information.

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
