# Peer review of "Reconstruction of super-resolution from high-resolution remote sensing images based on convolutional neural networks"

_PeerJ Computer Science, doi:10.7717/peerj-cs.2218_

## Round 0.1 · original submission · Major Revisions

Dear authors,

Thank you for submitting your article. Feedback from the reviewers is now available. It is not recommended that your article be published in its current format. However, we strongly recommend that you address the issues raised by the reviewers, especially those related to readability, experimental design and validity, and resubmit your paper after making the necessary changes.

Best wishes,

**Language Note:** The review process has identified that the English language must be improved. PeerJ can provide language editing services - please contact us at [email protected] for pricing (be sure to provide your manuscript number and title). Alternatively, you should make your own arrangements to improve the language quality and provide details in your response letter. – PeerJ Staff

Reviewer 1 ·

Basic reporting

The author proposed a novel algorithm named Edge-Enhanced Generative Adversarial Network in order to address the issues of edge fuzziness in remote sensing image super resolution. The experiments have been performed on two remote sensing data sets i.e. NWPUVHR-10 and UCAS-AOD which shows significant improvement in evaluation indexes.
The structure of the article is coherent. The introduction gives an understandable motivation for the need for action. However, many studies have been discussed in the literature. However, the references from [9 - 11] are very old. They need to be eliminated. Latest research articles addressing the matter should be cited. Please address the argument mentioned at Line # 70 which seems to be irrelevant at this point. The author has claimed that EGAN enhances edge details by integrating an edge enhancement network and improving mask branch for complex attention maps. How these issues have been addressed? From Line # [81 - 87], there is a repetition of text. Kindly eliminate repeated text. The reviewers highly recommend the authors to improve the language of the manuscript. On Line [123 – 124], the author mentioned that algorithm aims to achieve 4 times improvement in remote sensing image super-resolution reconstruction. How the claim has been achieved? On line [134 – 135], the author is relying on the arguments of conference papers to build the base of paper. In my view, to rely heavily on conference papers is a very weak step for research. The author has cited [13,14,15,17,18] as a conference papers (5 out of 18). To me, the referencing section seems to be very weak and more references needs to be added. On Line [162 – 163], it is mentioned that the proposed method replaces the upsampling used in the SRNet and EEN with sub-pixel convolution. Did you perform comparison for the selection of the said method? Any justification. What benefits have been achieved by performing the said task?
Why Adam optimizer was used? Why not others? On Line [355], it is mentioned that EGAN algorithm discussed in Chapter 2. Please eliminate this. On line [373], eliminate “proposed in the chapter”.

Experimental design

The paper and the experiment fit the aims and scope of the paper. However, there are few open questions remained. The authors claim that the algorithm achieve 4 times improvement in remote sensing images? How the claim has been achieved with respect to results?

Validity of the findings

The contribution can be enhanced by stating the theoretical contribution of the experimental findings. Lastly, the section conclusion and future work lacks a detailed description of future research, shortcomings and the contributions of the paper.
Was any Research Synthesis was performed which clearly indicate that the proposed model is outperforming existing research? Research Synthesis should be included to reflect contribution and outstanding performance of the proposed algorithm.

Annotated reviews are not available for download in order to protect the identity of reviewers who chose to remain anonymous.

Reviewer 2 ·

Basic reporting

This article provides a comprehensive exploration of super-resolution reconstruction of high-resolution remote sensing images using convolutional neural networks (CNNs). Below are some improvement comments to enhance the clarity, structure, and content of the article:

Ensure that the introduction clearly defines the problem statement and the significance of the research in the context of remote sensing and image processing.

Methodology Description
Overall the methods adopted seem to be fine and justified, but please provide a clearer explanation of the algorithms and techniques used in the proposed models (EGAN and ENDGAN). You also need to break down the complex technical terms and equations into simpler language for better understanding by readers who may not be experts in the remote sensing and imagery field.

A picture is more worth than words so, use more illustrative figures and diagrams to visually represent the architecture and workflow of the proposed models.

Experimental design

Experimental Setup and Evaluation:
Provide more details about the experimental setup, including hyperparameters used, data preprocessing steps, and training/validation/testing procedures. Also discuss in bit more details about the dataset

Clarify how the evaluation metrics (SSIM, PSNR, RMSE) were calculated and interpreted in the context of image quality assessment.

Validity of the findings

Validity of the research
Present the experimental results in a more organized and readable format, such as tables and graphs, to facilitate easier comparison between different algorithms and state of the art work.
Provide a more detailed discussion and interpretation of the results, highlighting the strengths and limitations of each algorithm and explaining the observed trends.
Summarize the key findings and contributions of the research more concisely in the conclusion section.
Provide insights into potential future research directions and areas for improvement based on the findings of the current study.

Additional comments

Language and Style:
Ensure consistency in terminology and writing style throughout the article.
Proofread the manuscript for grammatical errors, typos, and awkward phrasings to improve readability.

How do the introduced edge enhancement module and non-local module in EGAN and ENDGAN, respectively, compare to previous methodologies in terms of effectiveness and computational efficiency in processing remote sensing images?
Artifact Discrimination: What challenges does the artifact discrimination method in ENDGAN face when differentiating between artifacts and real details in super-resolution images, and how does it impact the overall image quality and authenticity?
Applicability and Scalability: Considering the advancements presented by EGAN and ENDGAN, what are the implications for the scalability of these methods in processing large datasets of remote sensing images, and how applicable are they to real-world scenarios beyond the tested datasets?

---

## Round 0.2 · accepted · Accept

Dear authors,

Thank you for the revision and for clearly addressing all the reviewers' comments. I confirm that the paper is improved. Your paper is now acceptable for publication in light of this revision.

Best wishes,

Reviewer 1 ·

Basic reporting

All comments have been addressed by the author.

Experimental design

All comments have been addressed by the author.

Validity of the findings

All comments have been addressed by the author.

Reviewer 2 ·

Basic reporting

The paper uses clear, professional English, providing adequate context and referencing relevant literature.

Experimental design

The study presents original research within the scope, addressing significant issues in super-resolution imaging. Methods are well-detailed, enabling replication, and the research is conducted to a high technical and ethical standard.

Validity of the findings

Findings are robust, statistically sound, and well-supported by data. Conclusions are linked to the research questions and supported by experimental results, demonstrating significant improvements over existing methods.

Additional comments

The manuscript is a valuable contribution with innovative solutions for edge enhancement and artifact discrimination. Overall, the study is recommended for acceptance.